# Integrating Socioeconomic Status and Spatial Factors to Improve the Accessibility of Community Care Resources Using Maximum-Equity Optimization of Supply Capacity Allocation

**DOI:** 10.3390/ijerph18105437

**Published:** 2021-05-19

**Authors:** Ming-Hseng Tseng, Hui-Ching Wu

**Affiliations:** 1Department of Medical Informatics, Chung Shan Medical University, Taichung 40201, Taiwan; mht@csmu.edu.tw; 2Department of Medical Sociology and Social Work, Chung Shan Medical University, Taichung 40201, Taiwan; 3Social Service Section, Chung Shan Medical University Hospital, Taichung 40201, Taiwan

**Keywords:** socioeconomic status (SES), measures of health inequality, community-based care access, accessibility, maximum equity, optimization, aging in place

## Abstract

Health promotion empowers people, communities, and societies to take charge of their own health and quality of life. To strengthen community-based support, increase resource accessibility, and achieve the ideal of aging, this study targets the question of maximum equity with minimum values, taking distances and spatial and non-spatial factors into consideration. To compare disparities in the accessibility of community care resources and the optimization of allocation, methods for community care resource capacity were examined. This study also investigates units based on basic statistical area (BSA) to improve the limitation of larger reference locations (administrative districts) that cannot reflect the exact locations of people. The results show the capacity redistribution of each service point within the same total capacity, and the proposed method brings the population distribution of each demand to the best accessibility. Finally, the grading system of assessing accessibility scarcity allows the government to effectively categorize the prior improvement areas to achieve maximum equity under the same amount of care resources. There are 2046 (47.26%) and 396 (9.15%) BSAs that should be improved before and after optimization, respectively. Therefore, integrating socioeconomic status and spatial factors to assess accessibility of community-based care resources could provide comprehensive consideration for equal allocation.

## 1. Introduction

### 1.1. Accessibility, Socioeconomic Status, Health Equity, and Aging in Place

Access to goods and services, the built environment, social norms, and other factors relevant to health are often determined by the community [1,2]. Accessibility refers to the ease with which relevant resources (such as medical care or social welfare) can be obtained from the location of population on demand. The accessibility assessment of community care resources can be determined using multiple factors, including spatial and non-spatial factors [3,4]. Because access to equitable community-based care is accepted as an important base for achieving the goal of aging in place, it might be thought that the requirement for community care stations is related to the population’s need. Therefore, integrating socioeconomic status and spatial factors to assess the accessibility of community-based care resources could enable part of the comprehensive considerations necessary for the equal distribution of governmental budgets.

Health promotion enables and empowers people, communities, and societies to take charge of their own health and quality of life. Socioeconomic status (SES) is usually measured by a certain level of education, income, or occupation or by a composite of these dimensions to create a total measure for a family or individual. Mueller and Parcel [5] defined SES as the relative position of a family or individual in a hierarchical social structure, based on the hierarchical social structure to access or control over wealth, prestige, and power. SES has been applied to a large concept, indicating that the position of persons, families, households, and census tracts or other aggregates established through these measurements to evaluate the equity of public policy. SES has also been taken to include measures of participation in social, cultural, or political life [6].

Investigating SES and its relationship to health and well-being is an important issue for economic inequality and to meet the demand for improving social reporting of relationships between SES and health, which often show evidence of the perseverance of differential health outcomes. Epidemiological and sociological analyses of health also generally use educational attainment as a measure for the SES. Some studies have indicated that income and occupational position have additional, distinct influences on health outcomes [7,8,9]. The association between educational attainment and seniors’ health often represents the one-way impact of SES on health, as educational attainment is often completed before the onset of adult health problems, and the association between income and health represents a bidirectional influence [10]. The measure of SES on health in sociological and epidemiological studies is important for exploring the relationship between SES and risk factors for health [11].

Although occupational position is much more static than income over the life course, it may have a similar impact to that of educational attainment in the study of health. For some health outcomes, these two SES indicators do not have mutually exclusive associations with health [11]. According to the World Health Organization (WHO) definition of health literacy [12], this term “implies the achievement of a level of knowledge, personal skills and confidence to take action to improve personal and community health by changing personal lifestyles and living conditions.” Thus someone with this quality can read pamphlets and make appointments. Knowledge and understanding are powerful tools for health promotion. Health literacy is critical for empowering and improving people’s access to health information, as well as their capacity to use it effectively. Educational attainment could help people use a range of information and skills to make the right decisions concerning their health, so educational attainment may enhance health literacy. The WHO 2030 Agenda for Sustainable Development [13] indicates that health literacy is more than a personal resource: higher levels of health literacy within populations could yield social benefits as well. For example, higher levels of health literacy could, by mobilizing communities, address the social, economic, and environmental determinants of health. Studies have shown that individuals with low income and low education are often more likely to develop physical and mental health conditions due to their disadvantaged work environment and poor financial abilities [2,6,7,8,10,11,14]. Health inequities are endemic in every region of the world, and rates of disease are significantly higher among the poorest or most disadvantaged populations. These populations also suffer greater multidimensional costs of illness. This health inequality is not accidental, as the poor are more likely to live, work, study, and play in environments that are harmful to health. Individual and household poverty prevents disadvantaged people from having full access health resources. If vulnerable populations are empowered in early and sustained health promoting actions, this could prevent acute and chronic conditions or promote active and curative treatments.

For this reason, this study selects educational attainment and income level as non-spatial factors to represent SES impact, integrating non-spatial and spatial factors to calculate the accessibility of community care resources. Health literacy work in communities can reduce inequities in health and beyond; for this reason, the relationships between educational attainment, poverty, and health are an important issue for measuring health inequality.

### 1.2. Spatial and Non-Spatial Accessibility Measurement

In this study, a contextual approach is applied to the influence of SES on health, given that contextual measures of SES refer to characteristics of the individual’s environment. Contextual approaches typically involve ecological area measures and can also involve multilevel analyses. Contextual approaches to SES examine the social and economic conditions that affect all individuals who share a particular social environment.

Spatial accessibility (or geographical accessibility) focuses on the importance of geographical location and transportation distance between population demand and suppliers, so it is suitable for developing a spatial analysis with geographic information system. Traditional official policies assess the spatial accessibility of various resource allocations, which are often based on the ratio of the resource supply to the population demand within a certain administrative area. This method is easy to calculate but cannot truly show the spatial difference between resource accessibility and fairness. To consider the true characteristics of different geographical distributions between supply and demand, previous studies developed improvements in accessibility assessments based on the three variables of population demand, resource supply, and the transportation distance from the demand point to the supply point, such as in the gravity-based accessibility model [15], the two-step floating catchment area (2SFCA) method [16], and the enhanced two-step floating catchment area (E2SFCA) method [17]. Wang [4] used distance weighting and a supply scale to calculate the probability of selection from demand point to each supply point and proposed the three-step floating catchment area (3SFCA) method. In response to the increasing reliance on the evaluation of information from the internet for public decision-making behavior in the digital age, Wu et al. [18] proposed the enhanced three-stage floating catchment area (E3SFCA) method, which integrates supply capacity, distance decay, and a Google rating mechanism to calculate the probability of selection.

The variables that affect non-spatial accessibility include many demographic and socioeconomic factors, such as social status, cultural background, economic income, gender, age, race, and so on [4,19]. Wang and Luo [20] applied factor analysis to reorganize 11 demographic and socioeconomic index statistics into three major factors (socioeconomic weakness, socio-cultural barriers, and high medical needs). In this way, spatial and non-spatial accessibility factors were combined to define the missing primary areas of medical resources in the state of Illinois. McGrail and Humphreys [21] combined factor analysis with relative eigenvalues to assign weight to spatial and non-spatial accessibility indicators to evaluate the primary medical accessibility of the state of Victoria, Australia. Tang et al. [22] used the ratio of the size of elderly population at a certain demand point to the largest-sized elderly population at any demand point as the probability of medical demand to calculate an accessibility assessment of medical care for the elderly in Taipei, Taiwan. Using statistical data on fire incidence and the number of casualties, Xia et al. [23] calculated the relative proportion for each demand point, and the two probability values were then added together to propose a non-spatial dimension for fire-fighting demand indicators in Nanjing, China.

According to the Establish Community Care Station Implementation Plan of Taiwan [24], community care stations are called upon to provide at least three types of non-medical services, including home visits, phone calls, meal services, and health improvement care. In Taiwanese government open data from 2019 [25], women aged 65 or older were the main users of community care stations’ services, with ratios in the range of 62.84% to 72.04%. However, only 27.96% to 39.18% of males aged 65 or above used community care stations’ services (please see Table A1). The highest level of service utilization was “health improvement activity,” mainly used by women (72.04%). In in this study, we consider the population of women aged 65 or over as a non-spatial factor for those with higher need for community care resources.

To consider the importance of non-spatial factors to accessibility measurement, this study uses the relative proportions of three indicators, including high resource need (weighting by the population of women aged 65 or over), economic weakness (weighting by proportion of low-income households and low-middle-income households) and cultural weakness (weighting by educational attainment of completed senior high school or below), at each demand point to assess the impact of non-spatial dimensions on the evaluation results for the accessibility of community care resources.

### 1.3. Modifiable Areal Unit Problem (MAUP)

Most socioeconomic statistics are compiled according to administrative units (such as counties, towns, cities, and villages). However, they are affected by aggregation factors or the size and shape of the spatial unit. The resulting data are often generalized, which may lead to different conclusions in spatial analysis. The problem of spatial distribution variation caused by differences in spatial units during data collection is referred to as the “modifiable areal unit problem” (MAUP) [26].

Dewulf et al. [27] used three units for spatial analysis to explore the influence of MAUP on the spatial accessibility of primary medical care resources in Belgium. The results show that the use of different spatial analysis units can lead to significant differences in spatial accessibility. Cabrera-Barona et al. [28] used two different spatial analysis units to compare the spatial accessibility of primary medical care resources in Quito, Ecuador, finding no serious difference of MAUP.

When studying the socioeconomic and territorial level, if the original data are to be used for study, they should be the closest to the actual situation and reduce the probability of analytical errors. However, it is quite difficult to obtain detailed individual data for the consideration of the public for personal privacy. Therefore, many countries have established statistical area systems as the basic unit for collecting, summarizing, and publishing various types of data. Due to the need to protect privacy, they provide statistics in small or special areas to meet different needs. For example, Topologically Integrated Geographic Encoding and Referencing [29] is used by the United States Census Bureau, and statistical area classification (SAC) [30] is used in Canada. The spatial units of Taiwanese geographical statistical classification [31] include the basic statistical area (BSA), the first-level dissemination area, the second-level dissemination area, the village/township, the city/county, regional planning, and the nation.

Understanding the population distribution is an important part of social and economic study. Previously, when conducting resources accessibility assessments, the population demand was positioned at the geographical geometric center point of a certain spatial unit or the population-weighted center point, and only that spatial unit could be provided. This can be shown from the average reference position for that unit’s population. This kind of result often cannot truly reflect the exact position of the different population demand. For this reason, our study calculates the accessibility of community care resources based on two spatial units (village and BSA) to investigate the effect of MAUP.

### 1.4. Optimization of Community-Based Care Resources

The main goal of a community-based care policy in Taiwan to empower people and communities by improving health literacy and increasing access to healthcare resources. Establishing community care stations throughout can enhance the social participation of the seniors with better than sub-health status and reduce medical expenses. Thanks to these considerations, the evaluation of the geographical accessibility of community care stations can indicate problems in the equity of the allocation of resources and can become an important reference for policymakers.

Community-based care resources could include community care stations, daycare providers, long-term care institutions, and medical institutions that are located in communities and serve the people who live there. Studies using keywords such as “community care”, “elderly care”, and “geographical accessibility” have focused on the geographical accessibility in healthcare resources, and related analyses highlight the variables of distance, population demand, and number of medical resource suppliers [16,17,20,28,32,33,34,35,36,37], all of which influence the utilization of home and community-based services among recipients of long-term care in Taiwan [38], as well as the accessibility of institutional healthcare facilities for the elderly [39,40,41]. Study of care resources in non-medical communities focus on the business model of community-based care institutions [42] and the types of services provided to people with disabilities and the elderly [43]. In addition, local case studies can assess the integration of family support and community care [44] to evaluate and investigate the geographical accessibility of community care stations [45] and the demand and supply allocation of community-based elderly learning resources [46]. Tseng and Wu [47] investigated the optimization of the spatial allocation of community-based care resources, taking the maximum, the mean, and the minimum values of the distances into consideration. Three analytical allocation solutions for supply capability optimization were derived to compare disparities in geographical accessibility. This study calculated accessibility based only on spatial factors. Less work has been done on the optimization of community-based care resources integrating spatial and non-spatial factors.

Methods of spatial optimization are frequently used to improve the distribution and supply of medical service providers. Wang [4] compared methods of healthcare resource allocation optimization and found that solutions to classic location–allocation problems can be found in the optimal effectiveness of resource allocation. Tao et al. [40] sought to optimize the allocation of elder care facilities, using the current spatial distribution of the elderly population in Beijing, China, and a model developed to ensure maximum equity. Liu et al. [48] integrated a two-step floating catchment area (2SFCA) method and a potential model to assess a better search radius. This study demonstrated that 600 m is close to the real travel distance of the elderly in Xi’an, China. Wu, Tseng, and Lin [18] applied the three-step floating catchment area method and considered distance, hospital capacity, and Google ratings to produce results that are in better accordance with people’s decision-making behavior to assess rehabilitation resource allocation in the community.

In the study of accessibility in community care resources, among the comprehensive factors to be examined are both spatial (population demand, location of service points, number of service points, and distance decay factors) and non-spatial (socioeconomic status, including high resource need, economic weakness, and cultural weakness). This study produced an evaluation method for accessibility and set a target for maximum equity, from which the analytical optimal solutions for resource capacity allocation were derived. In this study, the analysis of the population demand was based on populations aged 65 and above in BSAs of Chiayi County in Taiwan. The analysis of the supply points for resources was based on the number of community care stations of Chiayi County in Taiwan. The results analyzed for BSAs in different townships were compiled and analyzed to determine the allocation of the community care stations in different townships and the disparities in resource accessibility for the population demand in Chiayi County.

This study examined current distributions of the population demand and the community care stations, as well as the accessibility of these stations-to-population demand.

This study explored the following issues:

Investigating means to ameliorating the problem of the exact position of different demand populations, comparing the difference levels of accessibility of community care resources based on two spatial units (village and BSA) to investigate the effect of MAUP.The equity assessment of accessibility to community care resources by integrating non-spatial (socioeconomic status and gender) and spatial factors to demonstrate inequalities of allocation.Taking the target of maximum equity with minimum values for distances and non-spatial factors into consideration, results of before and after optimization of community care capacity allocation were examined to compare disparities in accessibility score of community care resources.Follow-up improvements to policies were suggested based on the grading system of accessibility scarcity assessment were suggested.

## 2. Materials and Methods

### 2.1. Data Collection: Study Area and Datasets

According to the statistics produced by the Taiwanese government [49], the aging index is rising year by year, from 65.05 in 2009 to 127.80 in 2020. The old age dependency ratio and the ratio of those aged 65 or above to all population in Chiayi County are the highest in Taiwan. The aging index is the ratio of the ratio of those aged 65 or above population to the number of young people (aged 0 to 14). The greater the index, the more serious the aging situation. The aging index of Chiayi County is 226.15, the highest in Taiwan (please see Table A2). According to the WHO, a super-aged society is one where more than 20% of their total population is aged 65 and older. Those aged 65 and above are 20.34% of the population of Chiayi County, making it the only super-aged society county in Taiwan. Population density is the lowest in west Taiwan (262.38 people/km^2^), which means that there is lower accessibility for the elderly to access resources in Chiayi County because the wider regional distribution. Therefore, the equal allocation for community care resources is important for policy. For this reason, our study takes Chiayi County as an example for follow-up analysis.

Chiayi County includes 18 townships, 357 villages, and 4329 BSAs. It covers an area of 1903.637 km^2^. The total population is 499,481, and the population density is 262.38 (people/km^2^) at the end of 2020.

Information on the population aged 65 and above in BSAs was retrieved from the Social and Economic Database of the NGIS (National Geographic Information System) Social and Economic Information Service, Ministry of the Interior, released in December 2020 [49]. Information on community care stations was retrieved from the open data of the Social and Family Affairs Administration at the Ministry of Health and Welfare, published in 2020 on the ministry’s website for community care stations services [25]. There are 140 stations in Chiayi County. Location data information indicate two stations at the same address. Therefore, there are 139 locations providing 140 community care stations’ services. This indicates that the entirety Chiayi County provides 140 units of capacity, of which 139 stations each provide one unit of capacity, and only one station provides two units of capacity.

Transportation is an important factor in seniors’ access to community care resources. Variability in the convenience of transportation among administrative districts includes differences in vehicle, shift frequencies, travel times, fare policies, and fare subsidy policies in counties/cities. Due to the scarcity or lack of credibility of the relevant data, it was not feasible to incorporate this information into the analysis of road network data. In the evaluation of the factors that affect geographical accessibility, this study drew from the research methods of Page et al. [50]. For cartographic data, numerical maps were drawn from the Ministry of Transportation and Communications [51]. To reduce possible error, road network data that represent actual route distances provided in government open data were adopted as the basis for the analysis of transportation-influencing factors, instead of traditional map distances (computed as the linear distance between two points). The software ArcGIS (Version 10.5.1, Esri, Redlands, CA, USA), which incorporates geographical information systems, was used to calculate geographical accessibility together with network analyses.

According to the Establish Community Care Station Implementation Plan of Taiwan [24], community care stations must provide at least three types of non-medical services, including home visits, phone calls, meal services, and health improvement activities. Thanks to the availability of care in the community, seniors can engage more closely with society and continue to live in familiar environments. The target group encompasses nearly all residents aged 65 or above whether healthy, with sub-health status, or disabled in need of home care. The estimation of the population demand for community care stations in this study was used to refine data analyses to 4320 BSAs in Chiayi County as the basis for statistical stratification. Next, BSAs belonging to 18 townships were compiled and analyzed to establish the allocation of community care stations in different townships. Finally, disparities were seen in the nearest resource accessibility of the population demand in the BSAs.

### 2.2. Integrating SES and Spatial Factors to Measure Accessibility Using Analytical Solutions for Optimization of Supply Capacity Allocation

The use of social welfare resources typically implies that the user has searched and selected a service provider within the available choices designated by policies and regulations due as constrained by the limitations of government finances and resources. In the present policy environment of Taiwan, resources provided by community care stations within a county/city can only be used by the residents of that county/city, and each user can only visit one service point for a given service. The Taiwanese government currently has rules in effect for the establishment of community care resources (for example, restricting service targets and service items), but it does not clearly regulate the area of the stations, service items, or number of people to be served. The question of appropriate service capacity could help distribute public resources more equally, but this has been little studied. For this reason, our study proposed an analytical approximation to integrating SES and spatial factors to access the accessibility of community care resources to optimize the supply capacity allocation. The current study assessed the optimization of accessibility to community care resources in favor of maximum equity given the limitation of total capacities. The research framework is shown in Figure 1.

We propose analytical solutions for the optimization of supply capacity allocation to calculate the accessibility score by considering spatial and non-spatial dimensions, as follows.

The national policy of Taiwan stipulates that the resources provided by community care stations within a county or city can only be used by the residents of that locality, and each user can only visit one service point for a given service. Wu and Tseng [45] and Tseng and Wu [47] proposed the nearest-neighbor two-step floating catchment area (NN2SFCA) method, as shown in Equations (1)–(3).
(1)$$A_{i}=\underset{j}{\sum }\frac{S_{j}*I\left(j\right)*f\left(d_{ij}\right)}{\sum_{k}P_{k}*I\left(j\right)*f\left(d_{jk}\right)}$$
(2)$$I\left(j\right)=\left\{\begin{array}{c} 1,\;j=j_{NN}\;for\;each\;i\\ 0,\;j\neq j_{NN}\;for\;each\;i \end{array}\right\}$$
(3)$$f\left(d_{ij}\right)=\left\{\begin{array}{c} 1,\;d_{ij}=d_{ij_{NN}}\leq 3\;\mathrm{km}\\ \frac{3}{d_{ij}}\;d_{ij}=d_{ij_{NN}}>3\;\mathrm{km}\\ 0,\;d_{ij}\neq \;d_{ij_{NN}}\; \end{array}\right\}$$
where *A_i_* is the geographical accessibility score for demand *i* and represents the average amount of supply point resources enjoyed by the population demand at the demand location *i*. *S_j_* represents the supply capacity for the point *j*. *P_k_* represents the size of the elderly population in the location at demand *k*. *I*(*j*) is a nearest-neighbor limiter, and *j_NN_* is the specific service point *j* found for the location at demand *i* in a nearest-neighbor search. *f*(*d_ij_*) is the distance decay function, and the search radius for resources in this study is divided into two zones in relation to the respective distances. The first zone ($d_{ij}\leq 3\;\mathrm{km}$) includes a range of points that an elderly person can reach by foot within one hour [45]. The second zone ($d_{ij}>3\;\mathrm{km}$) includes the range of points that an elderly person can reach on foot in one hour or longer.

To consider integrating SES and spatial factors to access accessibility of community care resources, the enhanced nearest-neighbor two-step floating catchment area (ENN2SFCA) method is proposed in this study, as shown in Equation (4).
(4)$$A_{i}=\underset{j}{\sum }\frac{S_{j}*I\left(j\right)*f\left(d_{ij}\right)}{\sum_{k}P_{k}*\beta_{i}*I\left(j\right)*f\left(d_{jk}\right)}$$
where $\beta_{i}$ represents the adjustment factor that considers the non-spatial dimension of the population demand. Resource need is evaluated by the ratio of population of women aged 65 or above at each demand point (called “*R*”), economic weakness is measured by the ratio of number of low-income households or low-middle-income households at each demand point (called “*E*”), and cultural weakness is evaluated by the ratio of population with an educational level of senior high school or below at each demand point (called “*C*”). As shown in Equation (5), the value of $\beta_{i}$ is between 0 to 3.
(5)$$\beta_{i}=\frac{R_{i}}{Max\left(R_{i}\right)}+\frac{E_{i}}{Max\left(E_{i}\right)}+\frac{C_{i}}{Max\left(C_{i}\right)}$$

Equation (5) indicates that the greater the relative proportion of the elderly female population and the greater the resource need, the lower the relative proportion of the elderly female population and the lower resource need. The greater the relative proportion of the economic weakness and the greater the resource need, the lower the relative proportion of the economic weakness and the lower resource need. In addition, the higher the relative proportion of the cultural weakness and the greater the resource need, the lower the relative proportion of the cultural weakness, and the lower the resource need.

Based on maximum equity, given the limitation of total capacities, Tseng and Wu [47] developed analytical solutions for the optimization of supply capacity allocation to calculate the geographical accessibility score. In this study, an extended solution is calculated to optimize the supply capacity allocation ($S_{j}^{opt}$) to calculate the accessibility score by considering spatial and non-spatial dimensions, as shown in Equations (6)–(8).
(6)$$S_{j}^{min}=A_{e}*\underset{k}{\sum }P_{k}*\beta_{i}*f\left(d_{j_{NN}k}\right)*Min_{i}\left(\frac{1}{f\left(d_{ij_{NN}}\right)}\right)$$
(7)$$A_{e}=\frac{\sum_{j}S_{j}}{\sum_{i}P_{i}*\beta_{i}}=\frac{S}{P_{\beta }}$$
(8)$$S_{j}^{opt}=\frac{S}{\sum_{j}S_{j}^{min}}*S_{j}^{min}$$
where *A_e_* is the weighted average of accessibility, *S* is the total supply capacity, *P_i_* is the population of the demand point *i*, and $\beta_{i}$ represents the adjustment factor of population demand, taking the non-spatial factors into account. $P_{\beta }$ is the total modified population demand.

Table 1 shows the accessibility score evaluation equations used in methods A0–A2.

### 2.3. Indicators of Describing the Accessibility Distribution and Evaluating Equity of Access

To evaluate equity in accessibility to community care resources for the population demand in Chiayi County, this study uses the enhanced nearest-distance two-step floating catchment area method to calculate the accessibility score, as shown in Equations (1)–(8). To describe the resource distribution of the accessibility scores, this study applies the following statistical indicators: mean, median, standard deviation (SD), minimum (Min), maximum (Max), full range (Max–Min), and quintile (Q20, Q40, Q60, and Q80). To evaluate the equity of access to resources per population demand point, this study uses five indicators, including mean squared error (MSE), |Median–Mean|, coefficient of variation (CV), Gini coefficient, and the ratio of the top 20% and lowest 20%.

For inequality, the mean, median, and Gini coefficient are often applied [52]. The median is the middle number in a sorted list of numbers, with the same amount of numbers above and below it. It is sometimes used in place of the mean in cases with a sufficient number of outliers in the sequence that could skew the average. The full range (Max–Min) indicates the broad range of values, the wider range the more scattering and unequal distribution. The quintile (Q20, Q40, Q60, and Q80) method uses four cutting points to demonstrate the distribution of the two extremes, which could indicate the disparity between the top 20% and lowest 20%.

The first evaluation index for equity of access, MSE, is used to compare the severity of the difference of supply and demand by accessibility scores between the townships/BSAs and the overall average of Chiayi County. The MSE measures the average squared difference between estimated and target values. It represents the degree of disparity between the geographical accessibility scores in villages and the target value for maximum equity (average value of Chiayi County), where larger values represent larger disparities in resource distribution. MSE was also used to conduct a discrepancy evaluation for optimization of the geographical accessibility score of the locations of the population demand.

The absolute value for median minus mean (|Median–Mean|) is the second indicator of inequality used in this study. Values for |Median–Mean| that are closer to 0.0 indicate smaller differences in the distribution of accessibility and greater degrees of fairness.

The value of CV shows the extent of variability of the data in the sample in relation to the population mean, which represents the ratio of the SD to the mean. This is a useful statistic to use in comparing the degree of variation from one data series to another, even if the means are drastically different from one another. Ideally, if the value of CV is lower, this indicates the lower dispersion of the data. Therefore, it can represent the equality of the community care resource allocation.

The quintile method and Gini coefficient are often used to evaluate the income inequality [53]. The Gini coefficient was developed by Italian statistician Corrado Gini, using the Lorenz curve as a measure for the equality of income distribution within a society [54]. Values for the Gini coefficient range from 1 to 0, where 1 represents complete inequality in annual income distribution, and 0 represents complete equality. Generally speaking, a Gini coefficient below 0.2 indicates a highly equitable income distribution, values in the range 0.2–0.3 are equitable, those in the range 0.3–0.4 are bearable, those in 0.4–0.6 show serious inequality, and those above 0.6 indicate high inequality [55]. For Gini coefficient values above 0.6, the administrative authority is advised to be on the alert for excessive income inequality, as this may lead to social conflicts. For this reason, the Gini coefficient is also termed the inequality coefficient. The Gini coefficient can indicate the seriousness of the distribution of income inequality. The greater the distance between the curve and the diagonal, the larger the coefficient, and the more uneven the distribution of income.

In this study, the quintile method was used to calculate the ratio of the top 20% to the lowest 20% to show the rich–poor gap. The distribution of community care resources is unequal in Taiwan, especially between urban and rural districts. The aging index of Chiayi County is the highest in Taiwan. This study therefore used these indicators to describe the accessibility distribution and evaluating equity of access of community care resources in Chiayi County. In summary, lower values for inequality indicators indicate better equity.

### 2.4. Grading System of Accessibility Scarcity Assessment

To achieve the maximum equity in the accessibility to community care resources, this study defines a grading system for areas in accordance with the concept of poverty line (as shown in Table 2). The poverty rate is the ratio of the number of people (in a given age group) whose income falls below the poverty line; taken as half the median household income of the total population. It is also available by broad age group: child poverty (0–17 years old), working-age poverty and elderly poverty (65 years old or above) [56]. Scarcity of community care resources results from the unequal allocation of public resources. In Taiwan, the poverty line is defined by determining whether the monthly income for each person in the household falls below the lowest living index, using this index to determine who is eligible for economic help, assistance during an emergency or disaster, and support in living independently [56]. Because accessibility of community care resources is an important support network for household and then achievement of aging in place, this study adopts the principle of the “lowest living index” to grade the scarcity of community care resources.

First, based on the BSAs of Chiayi County, the integration of spatial and non-spatial factors is used to calculate the accessibility scores of community care resources, and the median (=1.601) is obtained. The value of 60% of the median is defined as the low accessibility index (or the accessibility poverty line). Second, the value of 90% of the median is defined as the middle-low accessibility index. Third, the total number of resource supply in Chiayi County (total number of community care stations is 140) divided by the total number of people at demand (total number of people over 65 in demand is 92,160) is defined as the average accessibility index (140 ÷ 92160 = 1.519). Finally, the accessibility scores of community care resources of each demand point (the BSAs) in Chiayi County are divided into the following four levels based on the above three indices, as shown in Table 2.

Policy improvement suggestions can be identified in accordance with the grading system to redistribute the public budget. For example, the first-level of low resource areas can be listed as priority areas, in need of urgent improvement. Second-level areas can be given as moderate scarcity of resources improvement areas, and the third-level areas relate to mild scarcity of resources. These two grades’ areas can be listed as long-term improvement goals. In the fourth-level areas, the government can check whether there is an oversupply of resources.

## 3. Results

This study proposed inequality indicators and a grading system for accessibility scarcity assessment to evaluate equity in community care resources. First, the distribution of population demand and community care resources is presented. Second, to compare the impact of the MAUP on assessment of accessibility of community care resources, the spatial scales of “village” and “BSA” are used. Third, the equity of community care resource accessibility is compared before optimization with only geographical accessibility considered (method A0) and by integrating spatial and non-spatial dimensions (method A1). Fourth, for the goal of the maximum-equity optimization of supply capacity allocation, a comparison of two methods of calculating accessibility values is undertaken, based on the BSA level in Chiayi County, using analytical approximate solutions that assess the degree of inequality of community care resources before optimization (method A1) and after it (method A2).

### 3.1. Distribution of Population Demand and Community Care Resources

Table 3 provides an overview of resources of Chiayi County using the regional average method. For the over-65 population demand in townships, the accessible rates of service points per thousand elderly ranged between 0.579 and 2.888, and the accessible rate per thousand elderly of Chiayi County was 1.519. The accessibility rates of eight townships were lower than the average for Chiayi County. The accessibility rates of service points in the villages ranged between 8.333% and 78.261%, while the average value for the whole county was 39.216%. The accessibility rates of 11 townships were lower than the average value for the whole island. It is worth noting that in Chiayi County, which had the highest degree of urbanization and the highest density of elderly population, accessibility rates were lower than the average value for the whole island. For example, the accessibility rate for community care stations of per thousand elderly in Budai Township was the lowest, at 0.579, which means per thousand elderly had access to six community care stations. Another example is Alishan Township, which is located in a mountainous area and has 12 villages, with only one community care station. The accessibility rates of community care stations of villages in Alishan Township was the lowest (8.333) in Chiayi County. These data indicate that equity in the accessibility rates of community care stations for the elderly populations in Chiayi County leaves much to be desired.

### 3.2. The Impact of the MAUP on Assessment of Accessibility of Community Care Resources

In the evaluation of spatial accessibility, it is usually impossible to obtain the exact geographical location for each demand population. The most feasible way of doing this is to locate the population at the geometric center point or the population-weighted center point of a certain spatial unit. Using this method can only provide an average reference position for the population, but it cannot reflect the true position of the different demanders. In this study, we use two spatial units (villages and BSAs) to calculate geographical accessibility scores and explore the degree of influence of the BSAs on the evaluation of the accessibility of community care resources.

Table 4 presents the results of the assessment of the geographical accessibility of community care stations in Chiayi County on different spatial scales. The largest spatial unit is at the village level, and the smallest spatial unit is at the BSAs level. Chiayi County has 18 townships, 357 villages, and 4329 BSAs. The results show that the distribution of spatial accessibility values are all larger for the mean, median, SD, Max, and full distance (Range) because the spatial scale is smaller. The value of Min falls with the spatial scale from villages to BSAs. Comparing the inequality indicators at the village and BSA levels, such as MSE, |Median–Mean|, the CV, Gini coefficient, and the ratio of top 20% to lowest 20%, shows that the values for all of these indicators will become larger as the spatial scale is refined.

Table 4 shows that the impact of the MAUP, that is to say, the more detailed spatial units are used to present the average reference position of population demand, the distribution of spatial accessibility values of community care stations in Chiayi County will be wider. As the distribution becomes higher and wider, the resource allocation shows an unequal trend, and it becomes easier to assess which areas are showing a shortage of resources or excessively high resource distribution.

Figure 2 visualizes the unequal distribution of geographical accessibility values according to spatial scale units of villages and BSAs. As shown in Figure 2b, the lower accessibility areas (red areas) are increased when spatial scale refined to BSAs. This figure indicates the impact of the MAUP. 

### 3.3. The Equity Assessment of Community Care Resources Accessibility by Integrating Non-Spatial and Spatial Factors

The assessment of resource accessibility assessment can be determined using multiple factors; therefore, both spatial and non-spatial dimensions should be considered. This study uses three socioeconomic indicators (high resource need, economic weakness, and cultural weakness) to establish the impact of non-spatial factors on community care resources. Table 5 shows the degree of influence of spatial and non-spatial factors to the accessibility assessment results based on BSAs. The results show that when the spatial and non-spatial dimensions are simultaneously considered in the accessibility calculation, the distributions of the values of the accessibility indicators (the full Max–Min range) were increased. The values of inequality indicators of accessibility distribution, such as MSE, |Median–Mean|, CV, Gini coefficient, and ratio of top 20% to lowest 20%, were also larger.

Table 5 shows the results when SES factors are taken into account in the assessment of the accessibility of community care stations in Chiayi County, which are very different from those where only spatial factors are considered. When the spatial and non-spatial factors are simultaneously considered for the accessibility analysis (method A1), accessibility values are higher, all values of inequality indicators of accessibility distribution are also larger. That is, an assessment method integrating non-spatial factors can indicate which areas have serious shortage of resources or excessively high resources allocation.

Figure 3 mapped differences based on BSAs level; the distributions showed that whether simultaneously consider the spatial and non-spatial dimensions of accessibility. As shown in Figure 3b, the higher-accessibility areas (light blue and dark blue areas) are decreased when considered spatial and non-spatial factors.

### 3.4. Maximum-Equity Optimization of Supply Capacity Allocation

Table 6 lists two methods of calculating accessibility values based on the BSA level in Chiayi County using analytical approximate solutions that assess the degree of inequality of community care resources before optimization (method A1) and after optimization (method A2).

This result shows that the four inequality indicators following optimization (method A2) are more equal than before (method A1). For example, the value for MSE before optimization was 19.105, and after optimization, it was 0.136. The value for | Median–Mean | before optimization was 0.891, and after optimization, it was 0.126. The CV before optimization was 1.739, and after optimization, it was 0.193. The Gini coefficient before optimization was 0.555, and after optimization, it was 0.072.

Comparing the statistical indicators for the accessibility values using the analytical approximate solutions listed in Table 6, the values of mean, SD, and full range (Range) are smaller after optimization. Only the value of median is larger after optimization, which indicates that the unequal allocation has improved.

Based on the above, Table 6 shows that when method A2 is used to perform the accessibility analysis after the resource capacity optimization, the accessibility values show a lower and narrow uniform distribution. Which means the method A2 could reduce the distribution gap of the accessibility scores of the population demand, and it is most helpful to reduce the unequal distribution of resources; that is, the accessibility of the population demand can obtain improved fairness.

In Table 7, the quintile method is used to compare the accessibility scores obtained through these two assessment methods, indicating improvements in resource allocation equity. Within the class interval from Q20 to Q80, the dispersions of resource distribution disparity are presented. The distribution area for method A1 was larger, method A2 rendered the same value of 2.031. This result indicates that if resource allocation assessment is carried out according to method A2, over 80% of the population demand will have consistent accessibility and achieve the maximum-equity goal. The ratio of the top 20% to the lowest 20% is 1 through method A2, which indicates equal distribution after the optimization of supplier resources.

Figure 4 mapped differences based on BSAs level; the distributions showed the before and after optimization by integrating spatial and non-spatial dimensions of accessibility simultaneously. The dark blue areas are increased after optimization in Figure 4b, which means the allocation is equal.

Table 8 presents the results of capacity optimization before optimization according to method A1 and after optimization according to method A2 in Chiayi County. The results point out that if the total amount of input resources is fixed at 140, 10 of the 18 townships must be increased in their supply capacity of community care resources, and 8 townships must be reduced. For example, if Puzi City currently provides 5000 h/per month subsidy from the government, the Chiayi County Government could increase the subsidy to 10,193 h/month to the Puzi City.

Table 8 shows the capacity redistribution within the same total capacity (140 community care stations). After optimization, according to method A2, there were 10 townships/cities’ whose capacity should be improved. The ratio of results of methods A2 and A1 could simplify the improvement order. For example, the priority district is Puzi City, which should be increased to 10.193 stations.

## 4. Discussion

For the reasons of fiscal restraint by the government and the increasing age of the population, the provision of social welfare resources may need to be reduced. Tseng and Wu [47] collected population demand, supply resources, and road network maps to calculate geographical accessibility, and they proposed analytical solutions to optimize supply capacity allocation using NN2SFCA method to solve the problems of fiscal restraint. In this study, spatial and non-spatial factors were integrated using the ENN2SFCA method, and a grading system of accessibility scarcity was proposed to show the supplier amelioration.

Dewulf et al. [27] showed that using different spatial analysis units indicated significant differences in spatial accessibility. Cabrera-Barona et al. [28] showed that there was no serious difference in MAUP. In this study, as shown in Table 4, the impact of MAUP is significant with reference to statistical distribution but less significant regarding equity of access. The spatial units are smaller the closer that the accessibility assessment approach the real condition, so the use of BSAs can provide precise measurement.

In Table 5, the integration spatial and non-spatial factors demonstrates the seriousness of neglecting SES factors. For example, assessing spatial and non-spatial factors simultaneously (method A1), the values for MSE, |Median–Mean|, CV, Gini coefficient, and ratio of top 20% to lowest 20% all become larger. This indicates an unequal allocation when taking a comprehensive consideration.

Policy improvement suggestions can be identified according to grades to re-allocation the community care resources. As indicated in Table 2, Grade 1 indicates severe scarcity of resource areas, a prior areas to be improved. Grades 2 and 3 are moderate scarcity of resources areas and mild scarcity of resources areas, respectively. Grade 4 represents abundant resources areas, which could maintain the original capacity or decreased to achieve the goal of maximum equity.

In Table 9, there are 2046 (47.26%) BSAs should be improved before optimization (method A1). That is to say, the community care resources distribution is very unequal. After optimization (method A2), only 396 (9.15%) BSAs were marked as needing improvement. This entails 1650 BSAs whose accessibility scores were higher than the average accessibility index and only 190 BSAs below the value of low accessibility index through the process of optimization (method A2).

Table 10 shows the improvement of the number of population demand after optimization. Before optimization (method A1), 52.75% of the population demand lived in resources scarcity BSAs. After optimization (method A2), there was only 8.28% of the population demand living in resource-scarcity BSAs. In all, the accessibility scores of 40,981 seniors were higher than the average accessibility index, obtained through the process of optimization (method A2).

The use of the grading system for the accessibility scarcity assessment could provide a clear full view for evaluating resources allocation. According to method A2, BSAs in Grade 1 should be preferentially supported with more resources, followed by BSAs in Grade 2 and Grade 3. Doing so would be a convenient way to the enhance accessibility of community-based care resources as part of improved government policy.

## 5. Conclusions

Active aging and successful aging are related to health equity. Seniors’ physical limitations entail that geographical accessibility affects their ability to take advantage of community care resources, and this is reflected in equity in the design of resource allocation policies. Equity of access to healthcare is a crucial element in health equity in community care policy. For this reason, our study uses the method of maximum-equity optimization to identify Chiayi County’s community care resource allocation.

Taking Taiwan′s current community care resources as an example, the policy recommendations from this study are as follows:

First, this study compensates for the limitations of previous studies, some of which took only spatial factors into consideration [40,47,57,58] or positioned population demand at the geometrically weighted center points of the population [18,47,48]. The limitations of exact locations of the population demand were addressed by using BSA data to solve problems of MAUP. Therefore, spatial distribution variation caused by differences in spatial units during data collection could be decreased.

Second, in the overall consideration of accessing community care resources, this study integrated spatial and non-spatial factors to evaluate the social and economic conditions, including high resource need (proportion of the population of women aged 65 or above), economic weakness (the disadvantaged economic households), and cultural weakness (the population with an educational level of senior high school or below). If only spatial factors are considered, this will result in an underestimation of the inequality of accessibility.

Third, adopting analytical solutions for the optimization of supply capacity allocation determined with the ENN2SFCA allowed this study to integrate spatial and non-spatial (SES) factors to determine how community care resource accessibility could be optimized in favor of maximum equity under a total capacity limitation. To strengthen community-based support, improving resource accessibility and achieve the ideal of aging in place, this study makes contributions to policy implementation through is use of method A2 (where the minimum distance is adopted as the approximate representation of distances between *j* and locations at demand *i* that rely on services to find the minimum value for resource optimization capacity) and analyzes units based on BSAs. The results of this study show that when the location of each service point is fixed and under the same total amount of input resources, method A2 brings the population distribution at each demand point to the best accessibility.

Finally, the grading system for accessibility scarcity assessment proposed by this study can help the government effectively categorize prior areas of improvement to achieve maximum equity under the same total amount of care resources.

Community-based care resources could include community care stations, daycare providers, long-term care institutions, and medical institutions, which are located in communities and serve the people who live there and can be understood as community-based care resources in a broad sense. This study only considers the accessibility of community care stations. The authors suggest that involving comprehensive institutions for seniors to age in place, such as daycare providers, long-term care institutions, and medical institutions, can further improve conditions in the future.

## Figures and Tables

**Figure 1 ijerph-18-05437-f001:**
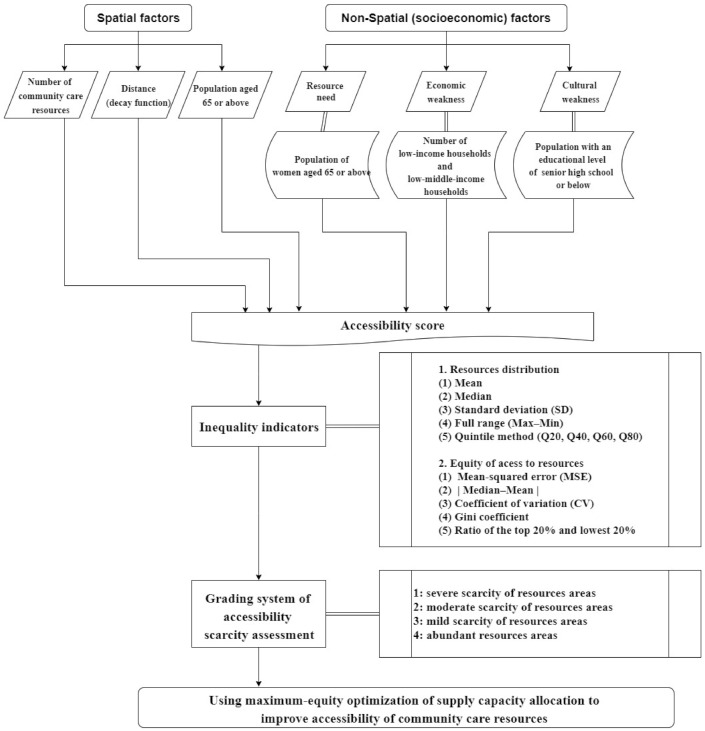
Research framework.

**Figure 2 ijerph-18-05437-f002:**
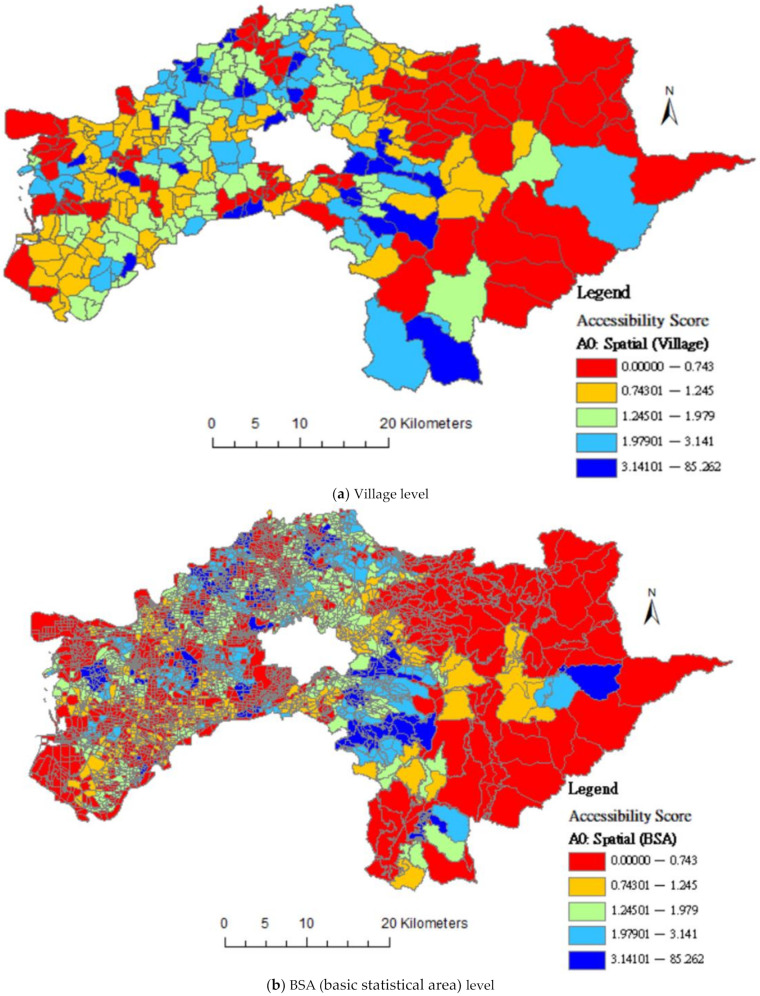
Differences in distribution of geographical accessibility score of community care resources by MAUP (Modifiable Areal Unit Problem).

**Figure 3 ijerph-18-05437-f003:**
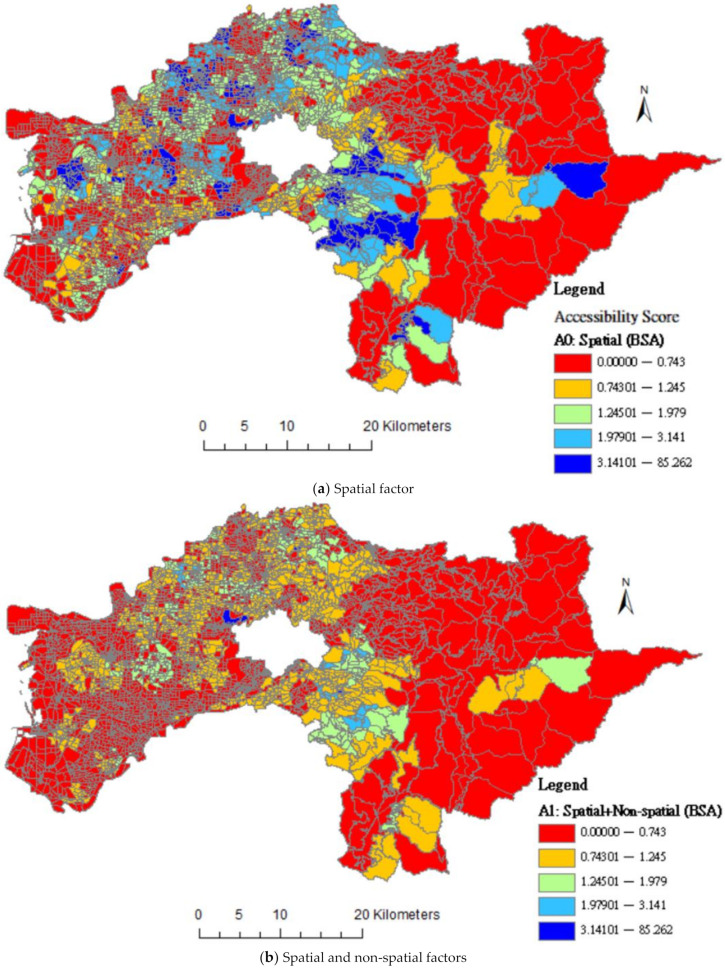
Differences in distributions of accessibility score of community care resources by spatial and integrating spatial and non-spatial factors.

**Figure 4 ijerph-18-05437-f004:**
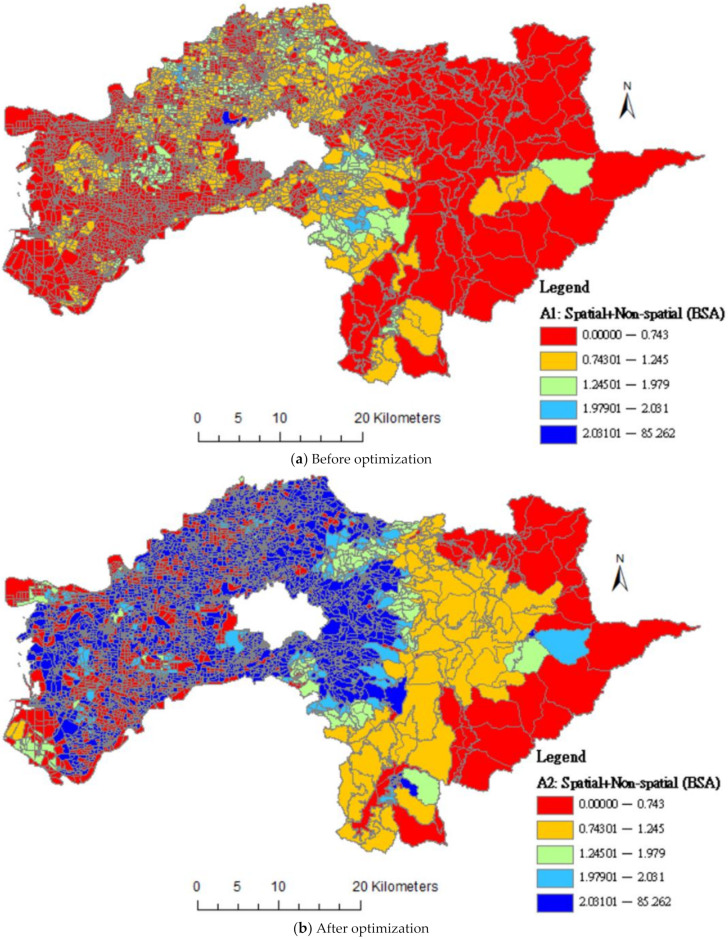
Differences in distributions of accessibility score of community care resources by before and after the optimization of supplier resources.

**Table 1 ijerph-18-05437-t001:** Definition of geographical accessibility calculation models.

Method	Description	Equation
A0	Geographical accessibility score—before the optimization of supplier resources	$A_{i}=\underset{j}{\sum }\frac{S_{j}*I\left(j\right)*f\left(d_{ij}\right)}{\sum_{k}P_{k}*I\left(j\right)*f\left(d_{jk}\right)}$ *S_j_* = 1 $I\left(j\right)=\left\{\begin{array}{c} 1,\;j=j_{NN}\;\;for\;each\;i\\ 0,\;j\neq j_{NN}\;\;for\;each\;i \end{array}\right\}$ $f\left(d_{ij}\right)=\left\{\begin{array}{c} 1,\;d_{ij}=d_{ij_{NN}}\leq 3\;\mathrm{km}\\ \frac{3}{d_{ij}}\;d_{ij}=d_{ij_{NN}}>3\;\mathrm{km}\\ 0,\;d_{ij}\neq \;d_{ij_{NN}}\; \end{array}\right\}$
A1	Accessibility score with spatial and non-spatial dimensions—before the optimization of supplier resources	$A_{i}=\underset{j}{\sum }\frac{S_{j}*I\left(j\right)*f\left(d_{ij}\right)}{\sum_{k}P_{k}*\beta_{i}*I\left(j\right)*f\left(d_{jk}\right)}$ *S_j_* = 1 $\beta_{i}\in \left[0,3\right]$
A2	Accessibility score with spatial and non-spatial dimensions—after the optimization of supplier resources	$A_{i}=\underset{j}{\sum }\frac{S_{j}^{opt}*I\left(j\right)*f\left(d_{ij}\right)}{\sum_{k}P_{k}*\beta_{i}*I\left(j\right)*f\left(d_{jk}\right)}$ $S_{j}^{min}=A_{e}*\underset{k}{\sum }P_{k}*\beta_{i}*f\left(d_{j_{NN}k}\right)*Min_{i}\left(\frac{1}{f\left(d_{ij_{NN}}\right)}\right)$ $A_{e}=\frac{\sum_{j}S_{j}}{\sum_{i}P_{i}*\beta_{i}}=\frac{S}{P_{\beta }}$ $S_{j}^{opt}=\frac{S}{\sum_{j}S_{j}^{min}}*S_{j}^{min}$

**Table 2 ijerph-18-05437-t002:** Grading system of accessibility scarcity assessment.

Grade	Description	Assessment Standard of Accessibility Values
1	severe scarcity of resource areas	below the value of “low accessibility index”
2	moderate scarcity of resource areas	between the values of “low accessibility index” and “middle-low accessibility index”
3	mild scarcity of resource areas	between the values of “middle-low accessibility index” and “average accessibility index”
4	abundant resource areas	equal or larger than the value of “average accessibility index”

**Table 3 ijerph-18-05437-t003:** Summary statistics of population aged 65 or above and community care stations, measured by administrative districts of Chiayi County.

Administrative District	Population Aged Over 65 Years	Percentage of Population Aged Over 65 Years	Number of Stations	Number of Villages	Stations-to Population (0/00)	Stations-to-Villages (%)
Taibao City	4854	5.267	8	18	1.648	44.444
Puzi City	7235	7.850	5	27	0.691 *	18.519 *
Budai Township	5180	5.621	3	23	0.579 *	13.043 *
Dalin Township	6211	6.739	13	21	2.093	61.905
Minxiong Township	10,317	11.195	18	28	1.745	64.286
Xikou Township	3181	3.452	4	14	1.257 *	28.571 *
Xingang Township	6233	6.763	18	23	2.888	78.261
Liujiao Township	5523	5.993	9	25	1.630	36.000 *
Dongshi Township	5315	5.767	8	23	1.505	34.783 *
Yizhu Township	4383	4.756	7	22	1.597	31.818 *
Lucao Township	3780	4.102	5	15	1.323 *	33.333 *
Shuishang Township	8098	8.787	8	26	0.988 *	30.769 *
Zhongpu Township	7046	7.645	15	22	2.129	68.182
Zhuqi Township	6683	7.252	6	24	0.898 *	25.000 *
Meishan Township	4133	4.485	3	18	0.726 *	16.667 *
Fanlu Township	2352	2.552	7	11	2.976	63.636
Dapu Township	841	0.913	2	5	2.378	40.000
Alishan Township	795	0.863	1	12	1.258 *	8.333 *
Total	92,160	100.000	140	357	1.519	39.216

Note. 1. * Lower than average. 2. This table was calculated based on the regional average method; however, in practice, the community care resources in Chiayi County can be used in the nearest or cross-townships.

**Table 4 ijerph-18-05437-t004:** Measures of geographical accessibility of community care resources by MAUP.

Method	Spatial Scale	Number	Mean	Median	SD	Range	MSE	|Median–Mean|	CV	Gini Coefficient	Ratio of Top 20% to Lowest 20%
A0	Village	357	1.455	1.118	1.227	8.870	1.509	0.337	0.843	0.416	3.413
A0	BSA	4329	1.623	1.229	1.668	27.725	2.794	0.394	1.028	0.462	3.703

Estimated by 1000 × capacity/people.

**Table 5 ijerph-18-05437-t005:** Measures of accessibility of community care resources by spatial vs. integrating spatial and non-spatial factors.

Method	Mean	Median	SD	Range	MSE	|Median–Mean|	CV	Gini Coefficient	Ratio of Top 20% to Lowest 20%
A0	1.623	1.229	1.668	27.725	2.794	0.394	1.028	0.462	3.703
A1	2.492	1.601	4.335	85.201	19.105	0.891	1.739	0.555	4.227

Estimated by 1000 × capacity/people.

**Table 6 ijerph-18-05437-t006:** Measures of accessibility of community care resources through methods A1 and A2.

Method	Mean	Median	SD	Range	MSE	|Median–Mean|	CV	Gini Coefficient
A1	2.492	1.601	4.335	85.201	19.105	0.891	1.739	0.555
A2	1.905	2.031	0.367	1.954	0.136	0.126	0.193	0.072

Estimated by 1000 × capacity/people.

**Table 7 ijerph-18-05437-t007:** Quintile accessibility of community care resources measured through methods A1 and A2.

Method	Q20	Q40	Q60	Q80	Ratio of Top 20% to Lowest 20%
A1	0.743	1.245	1.979	3.141	4.227
A2	2.031	2.031	2.031	2.031	1.000

Estimated by 1000 × capacity/people.

**Table 8 ijerph-18-05437-t008:** Assessment of supply capacity allocation of community care resources through methods A1 and A2.

Administrative District	Method		Ratio of A2/A1
	A1	A2	
Taibao City	8	7.029	0.879
Puzi City	5	10.193	2.039 *
Budai Township	3	4.713	1.571 *
Dalin Township	13	8.960	0.689
Minxiong Township	18	14.797	0.822
Xikou Township	4	4.240	1.060 *
Xingang Township	18	10.490	0.583
Liujiao Township	9	9.542	1.060 *
Dongshi Township	8	13.273	1.659 *
Yizhu Township	7	8.816	1.259 *
Lucao Township	5	7.927	1.585 *
Shuishang Township	8	12.242	1.530 *
Zhongpu Township	15	9.941	0.663
Zhuqi Township	6	9.200	1.533 *
Meishan Township	3	4.347	1.449 *
Fanlu Township	7	3.037	0.434
Dapu Township	2	0.909	0.455
Alishan Township	1	0.345	0.345
Total	140	140	1.000

Note. * Higher than original allocation.

**Table 9 ijerph-18-05437-t009:** Improvement of the number of BSAs by grading system of accessibility scarcity assessment through methods A1 and A2.

Method	Total	Numbers of BSAs				To Be Improved (Grades 1–3)	
		Grade 1	Grade 2	Grade 3	Grade 4	Subtotal	%
A1	4329	1223	739	84	2283	2046	47.26
A2	4329	190	163	43	3933	396	9.15

**Table 10 ijerph-18-05437-t010:** Improvement of the number of population by grading system of accessibility scarcity assessment through methods A1 and A2.

Method	Total	Numbers of Population				To Be Improved (Grades 1–3)	
		Grade 1	Grade 2	Grade 3	Grade 4	Subtotal	%
A1	92,160	30,195	16,777	1642	43,546	48,614	52.75
A2	92,160	3659	3131	843	84,527	7633	8.28

## Data Availability

Not applicable. This study applied governmental open data.

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
