# Peer review of "Integrating Socioeconomic Status and Spatial Factors to Improve the Accessibility of Community Care Resources Using Maximum-Equity Optimization of Supply Capacity Allocation"

_ijerph, 2021, doi:10.3390/ijerph18105437_

Round 1

Reviewer 1 Report

This is an interesting article on an important topic. At first I misunderstood the term "accessibility" with the concept of restricted mobility, but this was resolved when I read the paper. Perhaps it's better to clarify this in the abstract and beginning of the introduction section to avoid potential confusion/misunderstandings for others.

Furthermore, I have the following comments:

  • Table 1 would read better if it is put vertically
  • format of the paper seems to have a problem in some sections/sub-sections of it (there is too much indentation)
  • Is a standards charting solution used in Fig1? I am not an expert on this, but at the same time as a reader of the chart it's hard for me to understand what different shapes in the diagram mean?
  • A north arrow is missing in the maps of figure 2
  • While I understand there are some evaluations performed on the study, the authors have avoided to write a separate "section" or "sub-section" on "evaluation" of the results or evaluation of the method. I highly suggest the authors to dedicate a section to evaluation and provide detailed analyses on this. Thank you

Author Response

Dear reviewer,

Thanks very much.

Best regards,

Hui-Ching Wu

Reviewer 2 Report

Thais is a god paper, but I have some questions and suggestions.

line 46: Any references for the first sentence?
line 131-138: is it necessary all information of McGrail study? 7 lines?
line 130-146: again the same 7 lines to explain what other authors did????
line 147-154: same comments

Table 1, could be pass to a Supplementary Material, and one o two sentences with a summary of this table is enough.

line 186-200: is it necessary the detail that the authors explain what other countries do?

The authors use 6 pages of the paper for the introduction, from my point of view this section is too large and should be pruned.

Table 2 should be passed to the Supplementary Material, is not relevant for the paper.

Equation 5 should be rewrite in a easy read way, because is cut and in thre lines is very difficult ro read, may be using acronyms or similar.

Table 6. Not all the measures are necessary, Why do you put Min-Max and range together? An d the deviation of Mean and median? If the reader want to calculate is easy to do a subtraction. 
The same for table 7 and 8. Why do the authors use separate table (6,7,8)?? I can't understand.

I think that the majority of the tables can be put together (only 2 o three tables at the end) because the head is the same for some tabled.

About the the maps, are the results of this approach different from other spatial cluster analysis? Which are the advantages in the maps of the authors approach vs. other spatial cluster methods?  

Due to the difference extensions and demography in the difference study zones the figura 4  after optimization more realistic, should you compare using Disease Mapping methodology? 

Author Response

(The authors gave the same response as above.)

Round 2

Reviewer 1 Report

The revised version of the paper seems to address my raised concerns. Thank you.

Reviewer 2 Report

This new version imporve the older. Good job.